# IMD-mediated innate immune priming increases Drosophila survival and reduces pathogen transmission

Arun Prakash[1¤]*, Florence Fenner[1], Biswajit Shit[2], Tiina S. Salminen[3], Katy M. Monteith[1], Imroze Khan[2], Pedro F. Vale[1]

**1** Institute of Ecology and Evolution, School of Biological Sciences, University of Edinburgh, Edinburgh, United Kingdom, **2** Ashoka University, Sonepat, Haryana, India, **3** Faculty of Medicine and Health Technology, Tampere University, Tampere, Finland

¤ Current address: Vanderbilt University, MRB-III, Biological Sciences, Nashville, Tennessee, United States of America

* arun.prakash@vanderbilt.edu

**Data Availability Statement:** All raw data and code is available at 10.5281/zenodo.7624084. All other relevant data are in the manuscript and its supporting information files.

## Abstract

Invertebrates lack the immune machinery underlying vertebrate-like acquired immunity. However, in many insects past infection by the same pathogen can 'prime' the immune response, resulting in improved survival upon reinfection. Here, we investigated the mechanistic basis and epidemiological consequences of innate immune priming in the fruit fly *Drosophila melanogaster* when infected with the gram-negative bacterial pathogen *Providencia rettgeri*. We find that priming in response to *P. rettgeri* infection is a long-lasting and sexually dimorphic response. We further explore the epidemiological consequences of immune priming and find it has the potential to curtail pathogen transmission by reducing pathogen shedding and spread. The enhanced survival of individuals previously exposed to a non-lethal bacterial inoculum coincided with a transient decrease in bacterial loads, and we provide strong evidence that the effect of priming requires the IMD-responsive antimicrobial-peptide *Diptericin-B* in the fat body. Further, we show that while *Diptericin B* is the main effector of bacterial clearance, it is not sufficient for immune priming, which requires regulation of IMD by peptidoglycan recognition proteins. This work underscores the plasticity and complexity of invertebrate responses to infection, providing novel experimental evidence for the effects of innate immune priming on population-level epidemiological outcomes.

## Author summary

When we are vaccinated, our immune response is able to respond quickly if we are ever exposed to the same pathogen in the future. Unlike humans, the immune systems of invertebrates, such as insects, are not capable of the same type of specific immune memory. However, much work has shown that insects previously exposed to an inactivated pathogen will fare better if they are re-infected—a phenomenon broadly called "immune priming". How insects are able to do this is an exciting focus of current research. We investigated immune priming in the fruit fly, a powerful model system for infection and

**Funding:** We acknowledge funding and support from the Branco Weiss fellowship to PFV and a Darwin Trust PhD studentship to AP from the School of Biological Sciences, The University of Edinburgh, UK. The funders had no role in study design, data collection and analysis, decision to publish, or preparation of the manuscript.

**Competing interests:** The authors have declared that no competing interests exist.

immunity. We found that exposure to an inactivated form of the bacterial fly pathogen *Providencia rettgeri* resulted in flies better surviving a subsequent live infection. This effect lasted several days, was stronger in male flies, and was seen in different fly genetic backgrounds. We uncovered that priming requires a specific immune response in the fly fatbody (the equivalent to a fly 'liver') that produces an antimicrobial protein called *Diptericin*. We also found that primed flies were able to keep pathogen growth lower, and that this reduced their ability to spread the infection to other flies.

## Introduction

Immunisation using attenuated or inactivated pathogens is one of the most successful public health practices to reduce the incidence of infectious diseases [1]. Immunisation works because humans and other vertebrate animals have evolved an acquired immune response capable of specific immune memory, which ensures a strong, precise, and effective response to a secondary infection [2]. Insects possess a robust innate immune response to pathogens which includes both cellular and humoral components [3–5], but lack vertebrate-like specialized immune cells responsible for acquired immunity. These differences in immune physiology resulted in the long-standing assumption that invertebrates should not be capable of immune 'memory', though this view was clearly at odds with empirical evidence from several invertebrate host-pathogen systems [6–8]. A substantial body of work has revealed diverse priming responses in a range of arthropod taxa, including Dipterans: fruit flies [9], mosquitoes [10]; Coleopterans: flour beetles [11,12]; Lepidopterans: the greater wax-moth [13]; Hymenopterans: bumblebee [14]; Crustaceans: water fleas [15] and Arachnids: spiders and scorpions [16]. There is therefore substantial evidence that arthropods possess a form of "immune priming", where low doses of an infectious pathogen, or even an inactivated pathogen, can lead to increased survival upon reinfection.

There are many ways in which invertebrates may enhance their immune responses upon reinfection [6,17,18]. In *Drosophila*, the priming response during infection with the gram-positive bacterial pathogen *Streptococcus pneumoniae* was shown to be dependent on haemocytes and phagocytosis, while the Toll-pathway—the main pathway involved in clearance of gram-positive bacteria—was shown to be insufficient for successful priming [9]. Increased phagocytic activity in primed individuals has also been shown to play a key role in priming during *Pseudomonas aeruginosa* infection in Drosophila [19], while in the woodlouse, prior exposure to heat-killed bacteria led to increased phagocytosis by haemocytes upon reinfection [20]. In other *Drosophila* work, *PGRP-LB* (peptidoglycan recognition protein *LB*, a negative regulator of the Immune deficiency (IMD)-pathway) was identified as a key mediator of transgenerational immune priming against infection with parasitoid wasps (*Leptopilina heterotoma* and *Leptopilina victoriae*). Here, downregulation of *PGRP-LB* was necessary to increase haemocyte proliferation, required for wasp encapsulation by haemocytes in the offspring [21]. By contrast, in response to infection with *Drosophila C virus* (DCV), *Drosophila* progeny can produce a DCV-specific priming response by inheriting viral cDNA from the infected adult flies which is a partial copy of the virus genome [22,23]. The benefits of priming are not always associated with increased pathogen clearance, as shown during *Enterococcus faecalis* infection, where protection was explained by increased infection tolerance, not increased clearance [24].

Beyond its underlying mechanisms, an important but understudied aspect of immune priming is its potential consequences for pathogen transmission [17,25–27]. Epidemiological modelling predicts that priming should affect the likelihood of pathogen persistence, destabilise host–pathogen population dynamics, and that these effects depend on the degree of

protection conferred by priming [26]. Further theoretical work suggests that priming can either increase or decrease infection prevalence depending on the extent to which it affects the pathogen's colonization success and the host's ability to clear or tolerate the infection [25]. While primed individuals may live longer, thus extending the infectious period, their pathogen burden may be lower, which could lead to lower pathogen shedding and less severe epidemics. Immune priming is therefore expected to have a significant impact on the outcome of pathogen transmission by directly modifying important epidemiological parameters, but the strength and direction of these effects is not intuitive to predict.

Here, we focus on immune priming in *Drosophila* when infected with the gram-negative bacterial pathogen *Providencia rettgeri* to investigate the occurrence, generality, duration, and mechanistic basis of immune priming during systemic infection. Furthermore, motivated by the theoretical predictions about the role of immune priming on epidemiological dynamics, we also designed transmission experiments to enable us to test how immune priming could affect each of these behavioural and immunological components of pathogen spread.

## Results

### Immune priming in *Drosophila* is a long-lasting response, showing sex-specific effects in different genetic backgrounds

We first examined if the length of time between the initial non-lethal exposure with heat-killed *P. rettgeri* and the secondary systemic pathogenic challenge with live *P. rettgeri* affects the extent of priming. To address this, we exposed $w^{1118}$ male and female control flies to a systemic infection of live *P. rettgeri* 18-hours, 48-hours, 96-hours, 1-week or 2-weeks following the initial exposure to heat-killed bacteria. We found significant sex differences in the magnitude of the priming effect (sex treatment effect p<0.05) at 18-hours, 1-week and 2-weeks treatments (S1 Table), reflecting time-dependent and sex-specific priming dynamics. Male $w^{1118}$ flies showed increased survival after initial priming for time points 18-hours, 48-hours and 96-hours, and still showed a significant, albeit reduced, priming response 1-week and 2-weeks after the initial exposure (**Fig 1A–1E**). Female flies did not show a priming response when infected 18-hours following the initial challenge, but the priming response increased with 48-hours and 96-hours priming intervals before completely disappearing after a week time-interval (including 2-weeks) (**Fig 1A–1E** and **S1 Table**).

Next, we asked if priming occurred in two other commonly used *Drosophila* lines, Canton-S and Oregon-R (Ore-R) as observed with $w^{1118}$ (Fig 2A–2D). Given the widespread effects of the endosymbiont *Wolbachia* on *Drosophila* immunity [28–30] we also tested whether the presence of *Wolbachia* had any effect on immune priming by comparing the priming response of Oregon-R (OreR), a line originally infected with *Wolbachia* strain wMel, henceforth OreR-$^{Wol+}$ and a *Wolbachia*-free line OreR$^{Wol-}$ that was derived from OreR$^{Wol+}$ by antibiotic treatment [30]. Females and males of the four *Drosophila* lines were treated first with heat-killed *P. rettgeri*, followed by infection with a systemic infection with live *P. rettgeri* 96-hours after the first treatment. Since the 96-hour time gap between priming and live *P. rettgeri* treatments showed maximum priming response (difference in survival) for both males and females of the $w^{1118}$ line (Figs 1C and 2A), we kept the 96-hours timepoint as the consistent time-gap between priming and live infection throughout the study. w$^{1118}$ did not show differences between sexes in the priming responses at this timepoint (S2 Table; Sex × Treatment effect p = 0.93). Canton-S flies showed increased survival following priming and we observed a stronger survival benefit in females (Sex effect, p<0.001)(**Fig** 2C and **S2 Table**). Oregon-R males also exhibited increased survival following priming, but priming had no significant effect on the females, as indicated by a significant interaction between sex and priming

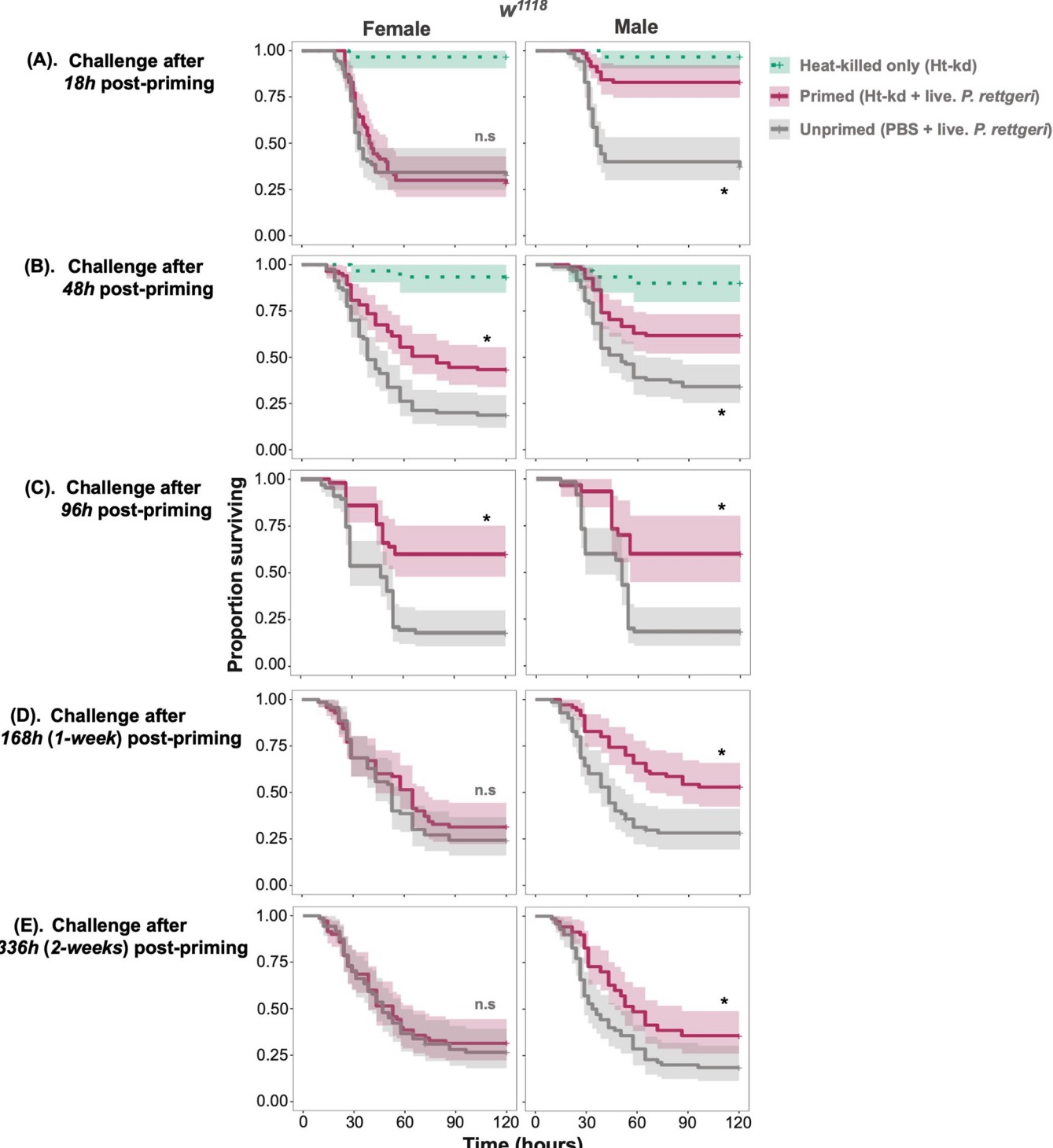

**Fig 1. The benefit of priming is long-lasting.** Survival curves of *w*<sup>1118</sup> females and males with primed (exposed to heat-killed *P. rettgeri* in the first exposure) and unprimed (exposed to a sterile solution in the first exposure) treatments challenged with live *P. rettgeri* pathogen after (A) 18-hours (B) 48-hours and (C) 96-hours (D) 168-hours/1-week and (E) 336-hours/2-weeks post priming that is, initial non-lethal exposure to heat-killed *P. rettgeri* (n = 7–9 vial with 10–15 flies in each vial/sex/treatment/timepoint).

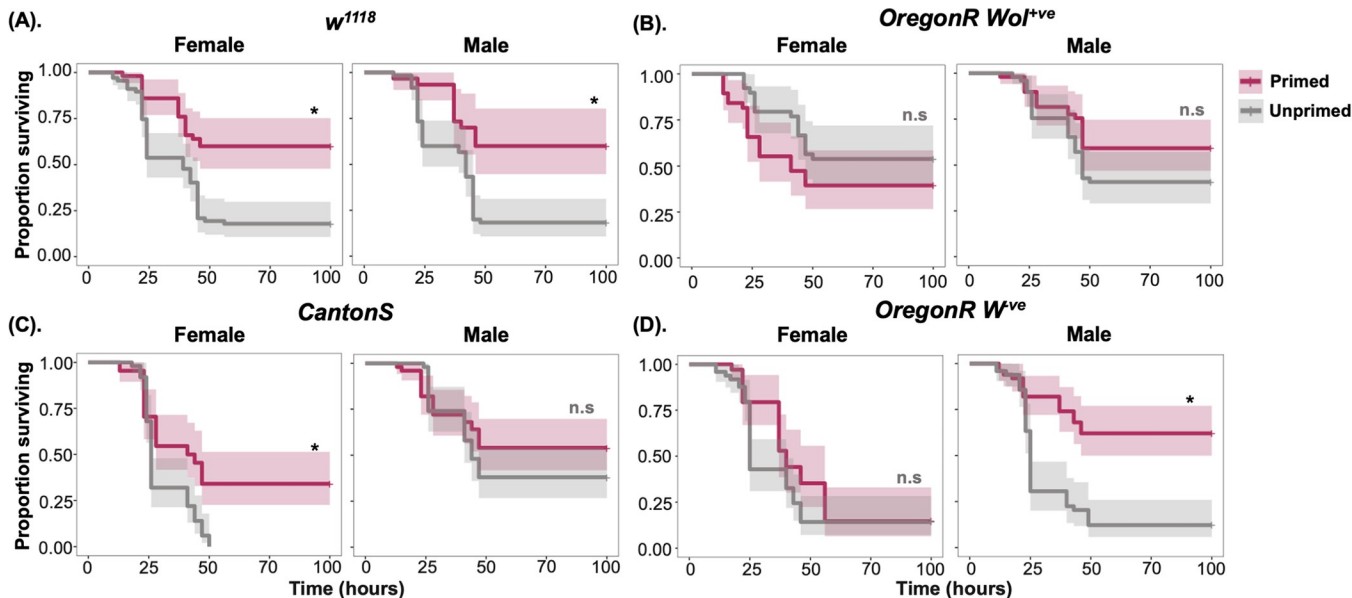

**Fig 2. Survival curves of females and males of the commonly used *Drosophila* lines either primed-challenged or unprimed-challenged with *P. rettgeri*.** (A) w[1118] –this is the same data plotted in Fig 1C, and included in this figure to allow a direct comparison with the other lines (B) Canton-S (C) Oregon-R carrying endosymbiont *Wolbachia* (D) Oregon-R cleared of *Wolbachia*. (n = 7–9 vials with 10–15 flies in each vial/sex/treatment/line).

treatment (Sex × Treatment effect p<0.001; **Fig** 2D and **S2 Table**). We found that the presence of *Wolbachia* significantly improved overall survival of both males and females (**Fig** 2C and 2D and S3 **Table**). However, the immune priming observed in males in absence of *Wolbachia* (OreR[Wol-]) was no longer present in flies carrying *Wolbachia* (OreR[Wol+]), and both sexes showed similar patterns of survival (Sex effect, p = 0.43; **Fig** 2B and S2 **Table**).

### Primed male *w[1118]* flies exhibit a transient reduction in bacterial load

Previous studies have shown that the priming response to fungi and gram-positive bacteria can be explained by increased clearance of bacterial pathogens in primed individuals [9,31]. We therefore examined whether the increased survival following a prior challenge we observed was a result of greater bacterial clearance in the primed individuals, or if the primed flies were simply better able to tolerate the bacterial infection [24]. To investigate this, we repeated the priming experiment with *w[1118]* as it showed the strongest priming phenotype in both sexes (**Figs** 1C and 3A), and measured the bacterial load at 24-hours and 72-hours post-infection for both primed and unprimed female and male flies. Again, we observed a clear priming effect in both sexes (**Fig** 3A and S4 **Table**). In primed males the systemic infection resulted in higher survival (Sex effect, p = 0<001), but the effect of priming on the survival benefit was the same in both sexes (Sex × Treatment effect p = 0.34; S4 Table) and also in decreased bacterial loads at 24-hours post-infection when compared to unprimed individuals (**Fig** 3 and S4 **Table** for survival and S5 **Table** for load). However, by 72-hours post-infection, bacterial loads had dropped in both primed and unprimed male flies and there was no detectable effect of priming on the bacterial loads (**Fig** 4B and S5 **Table**).

### Priming following oral infection reduces bacterial shedding and transmission in males

Apart from increasing survival during infections, immune priming may also have consequences to pathogen spread and transmission [27]. Despite having enhanced survival, primed

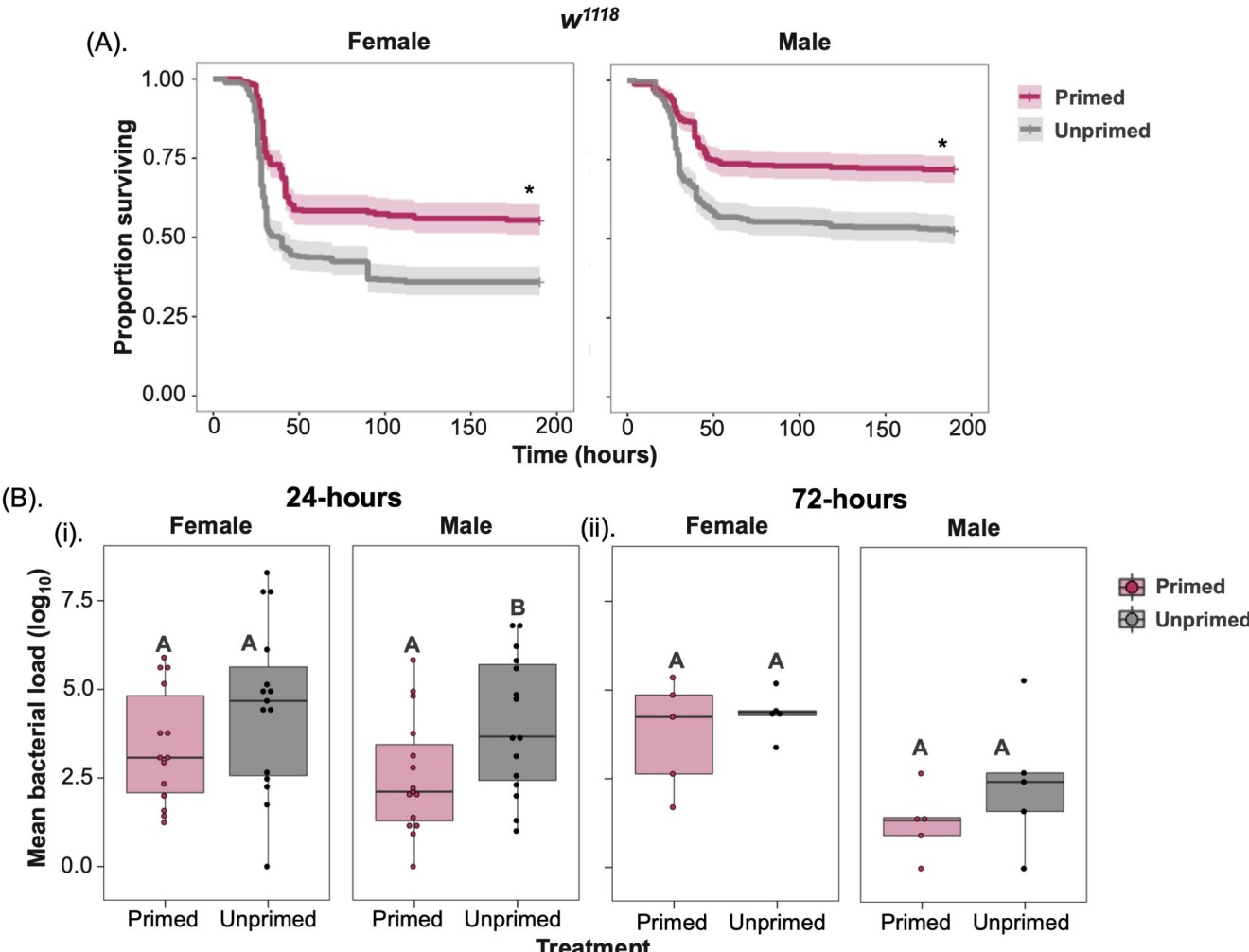

**Fig 3. Bacterial loads of primed and unprimed *w^1118* flies.** (A). Survival curves of primed and unprimed control *w^1118* flies (B). Internal bacterial load (n = 20–22 vial with 5–7 flies in each vial/sex/treatment) after (i) 24-hours and (ii) 72-hours following *P. rettgeri* infection. Different letters in panel-B denotes primed and unprimed individuals are significantly different, tested using Tukey's HSD pairwise comparisons, analysed separately for each timepoint and sex combination. The error bars in panel B represent standard error.

individuals may also extend the infectious period or their pathogen burden may be lower, which would lead to less severe epidemics [27]. To investigate how immune priming affects epidemiological parameters, we tested whether primed individuals have reduced bacterial shedding and spread. Given the oral-faecal nature of bacterial transmission and that our previous results all related to systemic infections, we first established that a survival benefit of priming also occurs under oral infection. Following an initial oral exposure to a heat-killed *P. rettgeri* culture, after a 72-hour period we exposed female and male *w^1118* flies orally to a lethal dose of live *P. rettgeri*. Primed *w^1118* males, but not females, showed increased survival decreased internal bacterial loads after priming via the oral route of infection. (**Fig** 4A and 4B and S6 and S7 **Tables**).

Given this priming effect on bacterial loads (Fig 4B) we hypothesised that priming could directly impact the amount of bacterial shedding, and therefore have a direct impact on pathogen transmission. We measured the shedding of single flies 4-hours after infection by live *P. rettgeri* exposure. We found that male flies previously primed with a heat-killed bacterial

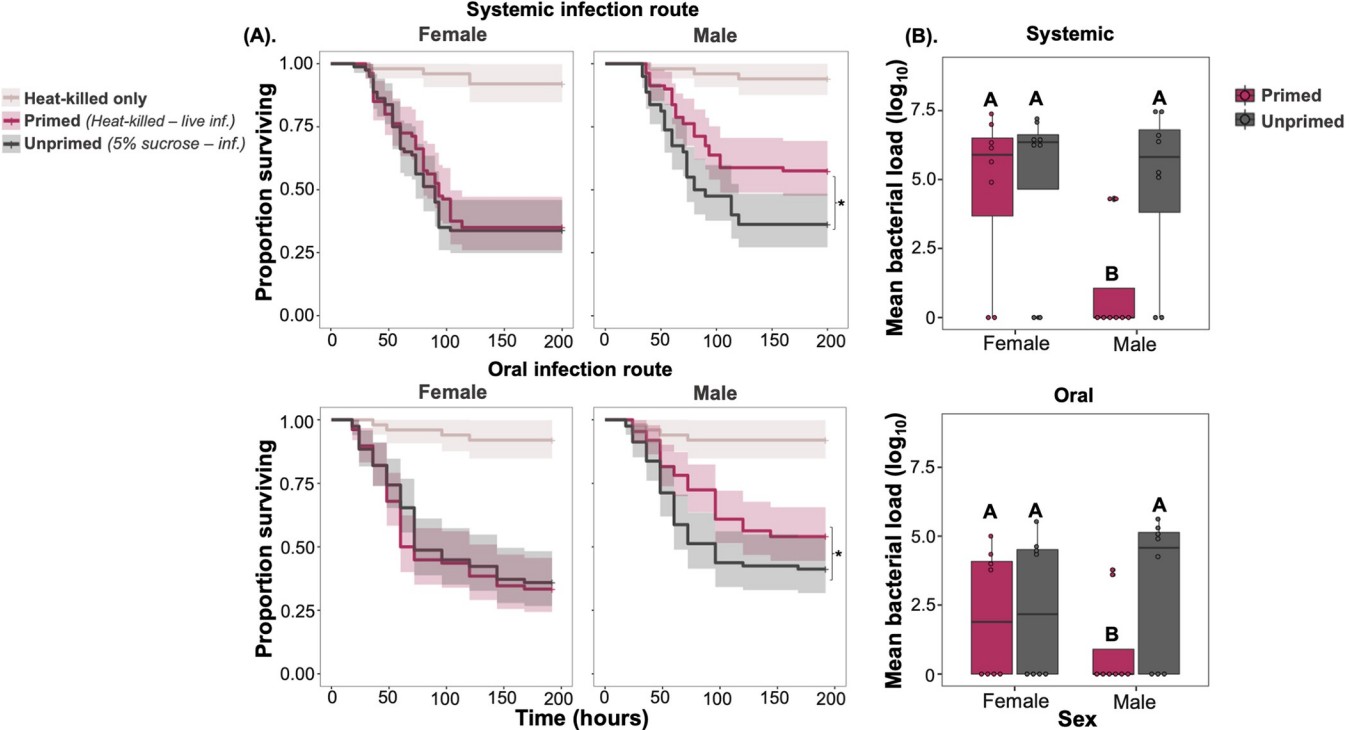

**Fig 4. Priming occurs in males but not females during oral infections.** (A) Survival curves of male and female wildtype ($w^{1118}$) flies with (i) systemic (ii) oral priming where flies initially exposed to heat-killed *P. rettgeri* or unprimed flies exposed to a sterile solution in the first exposure, this is followed by live *P. rettgeri* pathogen after 72-hour time gap [systemic infection dose = 0.75 OD (~45 cells/fly) and oral *P. rettgeri* infection dose = 25 $OD_{600}$]. (n = 7–9 vials/sex/treatment/infection route) (B). Mean bacterial load measured as colony forming units at 24 hours following (i) systemic and (ii) oral priming, followed by live *P. rettgeri* infection for male and female $w^{1118}$. Different letters in panel-B denotes primed and unprimed individuals are significantly different, tested using Tukey's HSD pairwise comparisons.

inoculum shed less bacteria than unprimed flies. However, both primed and unprimed females showed increased bacterial shedding following oral *P. rettgeri* infection (**Fig** 5A and S9 **Table**). These effects on bacterial shedding are likely to have a direct effect on the spread of infection in groups of individuals. In transmission assays, primed donor males also spread very little pathogen upon re-infection with oral *P. rettgeri*, compared to the unprimed male treatment where successful transmission was detected recipient flies (**Fig** 5B and S9 **Table**). However, both primed and unprimed females showed equivalent levels of bacterial spread following oral *P. rettgeri* infection (**Fig** 5B and S9 **Table**). While multiple traits, including host activity levels and contact rates may influence pathogen transmission [32,33], we did not find any difference in the locomotor activity of primed and unprimed flies (S3 Fig). Our results therefore provide evidence that, in male flies, immune priming reduces pathogen transmission by directly decreasing bacterial shedding from infectious flies.

## The *IMD*-pathway, but not the Toll pathway, is required for immune priming during *P. rettgeri* infections

In fruit flies, the production of AMPs during antibacterial immunity is mediated by the Immune deficiency (IMD) and Toll pathways. In both pathways, pathogens are recognised by peptidoglycan receptors (PGRPs), initiating a signalling cascade that culminates in the activation of the NF-κB-like transcription factors (*Dorsal* in Toll or *Relish* in IMD), resulting in the

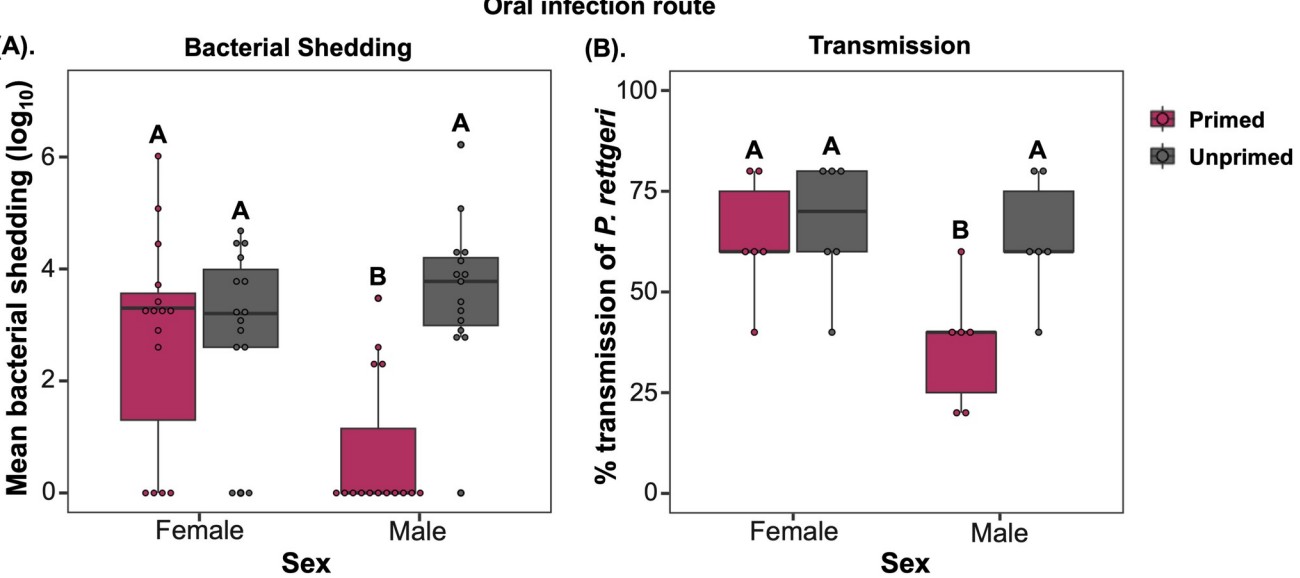

**Fig 5. The effects of priming on bacterial shedding and transmission.** (A). Bacterial shedding after 4-hours following oral priming (initial heat-killed exposure) and infection with live *P. rettgeri* (OD$_{600}$ = 25) for males and females (n = 15 individual flies per treatment). (B) percentage transmission of measurable bacterial loads for male and female *w*$^{1118}$ flies (recipient-flies) over the first 4-hours following exposure with infected donor flies (n = 6 independent transmission assays). Different letters denote primed and unprimed individuals are significantly different, tested using Tukey's HSD pairwise comparisons.

upregulation of AMP genes. The Toll-pathway generally recognises LYS-type peptidoglycan found in gram-positive bacteria and fungi. The IMD-pathway recognises DAP-type peptidoglycan found in gram-negative bacteria and produces AMPs such as *Diptericins*, *Attacins* and *Drosocin* among others. AMPs work with a high degree of specificity, so that only a small subset of the total AMP repertoire provides the most effective protection against specific pathogens [34], although this specificity has been shown to be greatly reduced during aging [35].

The inducible AMPs regulated by the IMD signalling pathway play a crucial role in resisting gram-negative bacterial infection such as *P. rettgeri* [36]. Therefore, we wanted to investigate whether the IMD signalling pathway and IMD-responsive AMPs contribute to immune priming. To address this, we used several transgenic fly lines (CRISPR knockouts and UAS-RNAi knockdowns) with functional absence of or knockdown of different regulatory and effector components of the IMD-signalling pathway. As all CRISPR/cas9 AMP mutants were isogenized onto the *iso-w*$^{1118}$ background, we first confirmed that the priming response in *iso-w*$^{1118}$ was the same as the *w*$^{1118}$ used in the previous experiments (Figs 6A and S4). First, we tested a *Relish* loss-of-function mutant *Rel*$^{E20}$, a key regulator of the IMD immune response. As expected, we found that *Relish* mutants did not show any bacterial clearance, died at faster rate, and therefore did not show any benefit of a priming treatment (**Fig** 6Aii for survival and **Fig** 6Bii for bacterial load, S10 **Table** for survival and S11 **Table** for bacterial load). Loss-of—function of *spätzle (spz)*, a key regulator in the *Toll* pathway, showed enhanced survival following initial heat-killed exposure, and their mortality rates were the same as in controls (*w*$^{1118}$) (**Fig** 6Aiii and S10 **Table**) indicating that Toll-pathway does not contribute to priming during *P. rettgeri* infection (**Fig** 6Aiii and S10 **Table**). In both lines where priming induced a survival benefit, the magnitude of effect was similar for males and females (Sex × Treatment effect >0.6; S10 Table).

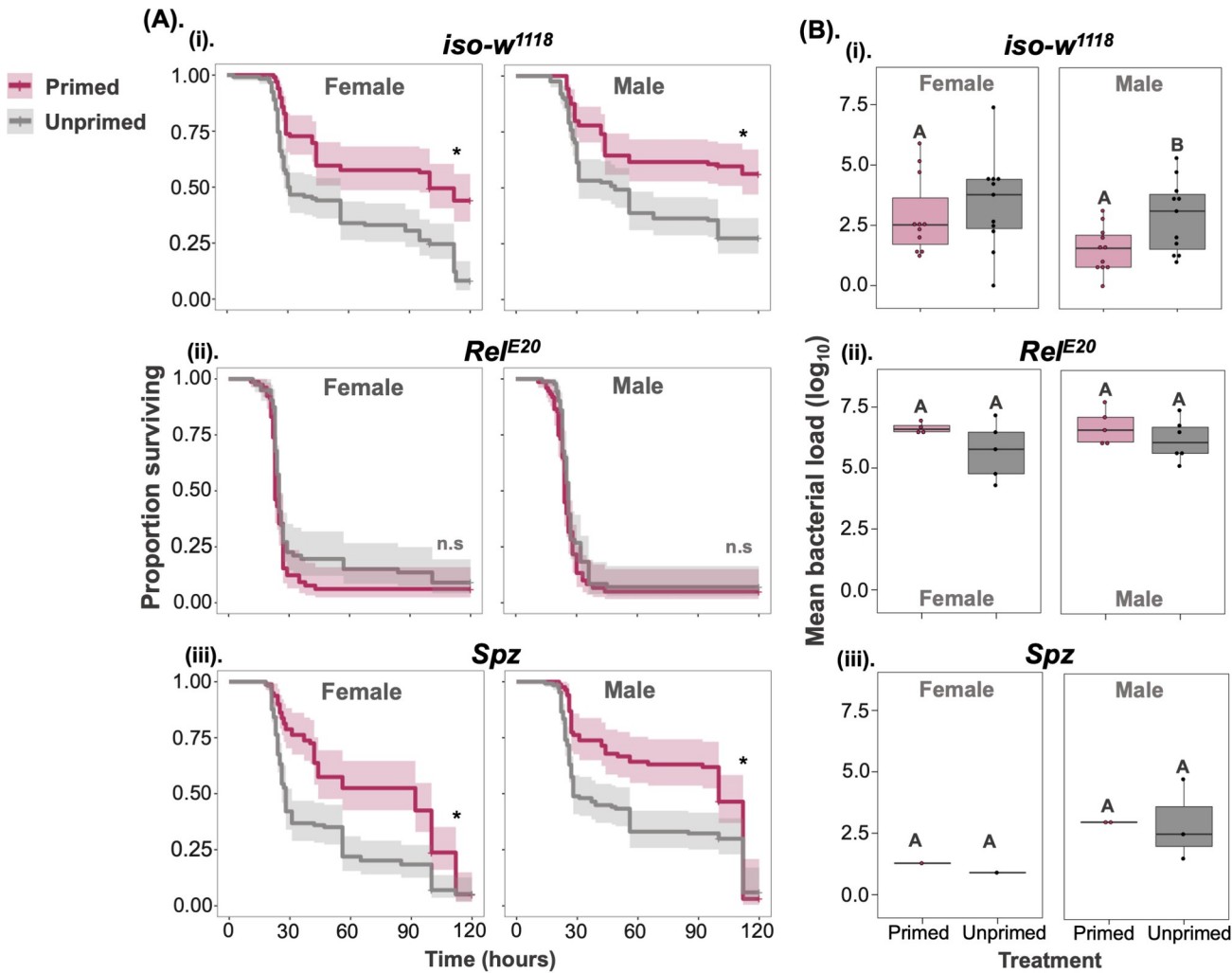

**Fig 6. Priming requires the IMD but not the Toll pathway.** (A): Survival curves for control flies and flies lacking different innate immune pathway components in females and males (i) control *iso-w^1118* (ii) *Rel^E20*- IMD-pathway transcription factor (iii) *Spz*- Toll pathway regulator in both males and females (B) (i-iii) bacterial load measured after 24-hours post-secondary pathogenic exposure (n = 7–9 vial/sex/treatment/transgenic lines). Different letters in panel-B indicate that primed and unprimed individuals are significantly different, tested using Tukey's HSD pairwise comparisons, analysed separately for each timepoint and sex combination. The error bars in panel B represent standard error.

## Expression of IMD-regulated *Diptericin B* in the fat body is required for priming

Given the important role of the IMD pathway for the priming response (Fig 6), next, we tested whether mutants with defective IMD signalling or unable to produce specific antimicrobial peptides were capable of immune priming. We first used a ΔAMP transgenic line, which lacks most of the known *Drosophila* AMPs (10 AMPs in total). ΔAMPs flies are extremely susceptible to the majority of microbial pathogens, including gram-negative bacteria [34], and we confirmed that both primed and unprimed ΔAMP flies succumb to death at a similar rate following systemic infection, and both primed and unprimed ΔAMP flies also exhibited elevated bacterial loads (measured 24-hours after the secondary pathogenic exposure) (**Fig** 7Ai for survival, **Fig** 7Bi for bacterial load, S10 and S11 **Tables**, relative to control w^1118 flies–compare with previous figure Fig 6Bi), showing that AMPs are needed for the priming response

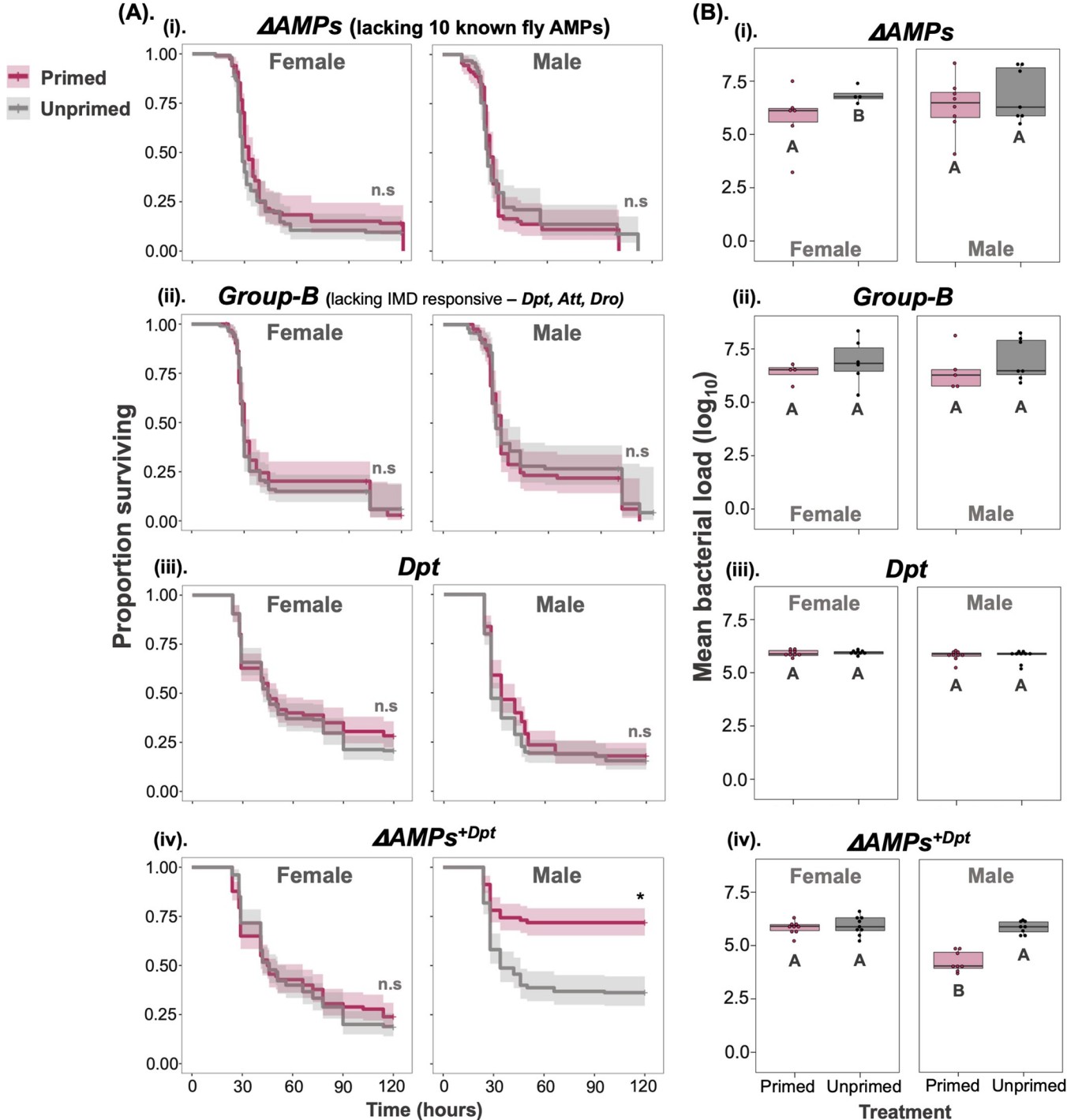

**Fig 7. Priming requires specific AMP expression.** (A). Survival curves for CRISPR/Cas9 AMP mutants (i) *ΔAMP* fly line: lacking all known 10 fly AMPs–(ii) *Group-B*: lacking major IMD pathway AMPs (iii) *Dpt* knockout and (iv) *ΔAMP^{+Dpt}*: lacking all AMPs except *Dpt*. (B) internal bacterial load quantified after 24-hours post-secondary exposure (n = 7–9 vials with 10–15 flies in each vial/sex/treatment/transgenic lines). Different letters in panel-B denotes primed and unprimed individuals are significantly different, tested using Tukey's HSD pairwise comparisons, analysed separately for each fly line and sex combination. The error bars in panel B represent standard error.

against *P. rettgeri*. To investigate which AMPs are required for priming against *P. rettgeri* infection, we used a Group-B transgenic line, lacking major IMD regulated AMPs (including *Diptericins* and *Attacins* and *Drosocin*) but have all upstream IMD signalling intact. Primed Group-B flies showed mortality similar to unprimed Group-B flies (**Fig** 7Aii and S11 **Table**) and exhibited increased bacterial loads across both the sexes irrespective of being primed or not (**Fig** 7Bii and S11 **Table**). Thus, despite being able to produce other AMPs, removal of IMD-regulated AMPs completely eliminated the priming effect, indicating that AMPs regulated by the IMD-pathway are required for immune priming against *P. rettgeri* infection.

Since *Diptericins* have been shown previously to play a key role in defence against *P. rettgeri*, we then used a *Dpt* mutant (lacking *Diptericin-A* and *B*) and $AMPs^{+Dpt}$ transgenic fly lines (flies lacking all known AMPs except *Diptericin*) to test whether *Diptericins* are required and sufficient for priming in both females and males. The survival benefit of priming disappeared in flies lacking $Dpt^{A+B}$ across both females and males and both primed and unprimed *Dpt* mutants exhibited increased bacterial load (**Fig** 7Biii and S11 **Table**). Notably, the priming response was recovered completely in male flies that lacked other AMPs but possessed functional *Diptericins* ($AMPs^{+Dpt}$). However, the same effect was not seen in females (Sex × Treatment effect $p<0.01$; **Fig** 7Aiv for survival, **Fig** 7Biv for bacterial load; S10 **Table** for survival and S11 **Table** for bacterial load).

In response to gram-positive *Streptococcus pneumoniae* infection, previous work described the role of haemocytes in immune priming through increased phagocytosis [9]. Subsequent work has also shown that reactive oxygen species (ROS) burst from haemocytes is important for immune priming during *Enterococcus faecalis* infection [37]. Since our results pointed to a *Diptericins* being required for immune priming against *P. rettgeri*, we wanted to determine if *Diptericin* expression in either the fat body or haemocytes was more important for immune priming. Using-tissue specific *Diptericin-B* UAS-RNAi knockdown, we found that male flies with knocked-down *DptB* in fat bodies no longer showed immune priming compared to the background or control iso-$w^{1118}$, while knocking down *DptB* in haemocytes resulted in a smaller but still significant increase in survival following an initial exposure (**Fig** 8 and S12 **Table**). This would therefore support that immune priming requires *DptB* expression in the fat body, while haemocyte-derived *DptB* is not important for priming. It is worth noting that it is possible that haemocytes contribute to immune priming through phagocytosis or melanisation, or via cross-talk with the IMD pathway, and this remains a question for future research.

## Priming is not an outcome of constant *Dpt* upregulation

As our results indicated that AMPs, especially *Diptericins* play a key role in *Drosophila* priming against *P. rettgeri*, we wanted to test if the priming effect we had observed was a result of constant upregulation of AMPs after initial heat-killed exposure allowing rapid bacterial clearance during the secondary exposure, or if AMP expression returned to a baseline level within the 96-hours between priming and the live infection. To do this, we measured *Dpt* gene expression at 18-hours and 72-hours following initial heat-killed exposure, and then 12-hours, 24-hours and 72-hours following secondary live *P. rettgeri* infection. *DptB* expression increased 18-hours after initial heat-killed exposure but returned to baseline levels by 72-hours (**Fig** 9A and 9B and S13 **Table**) indicating that flies did mount an immune response to the heat killed bacteria, but that this was resolved by the time they were infected with the live bacteria at 96-hours.

Further, 72-hours following a lethal secondary live *P. rettgeri* infection, we found that primed $w^{1118}$ females and males showed increased *DptB* levels compared to unprimed individuals (**Fig** 9A and 9B). The increased *DptB* expression also resulted in increased bacterial

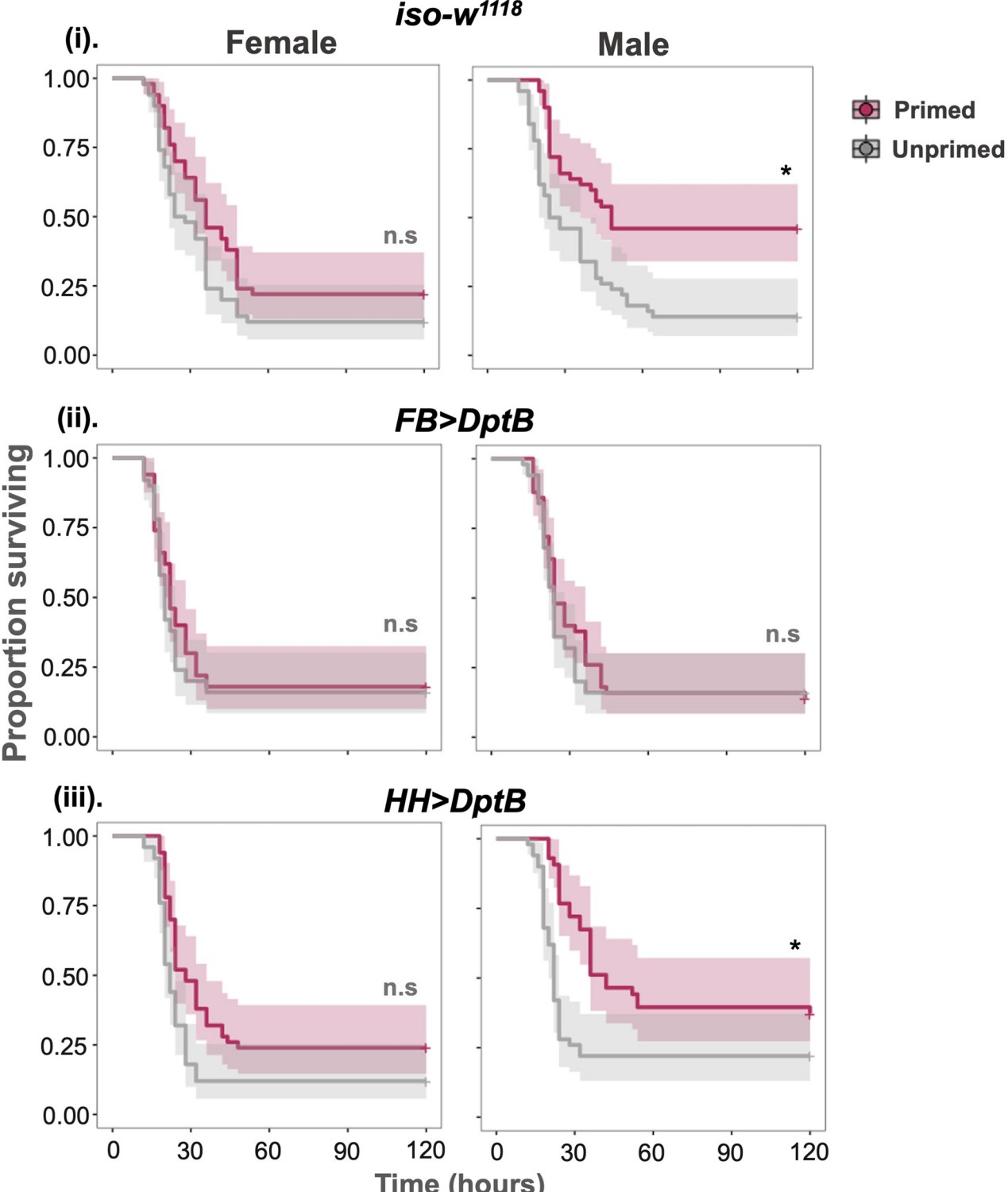

**Fig 8. Survival curves tissue-specific Dpt knockdowns.** (i) control *iso-w*[1118] and *UAS-RNAi* mutants—(ii) *FB>DptB* RNAi knock down of *DptB* in fat body and (iii) *FB>DptB* RNAi knock down of *DptB* in haemocytes. (n = ~50 flies/sex/treatment/fly lines).

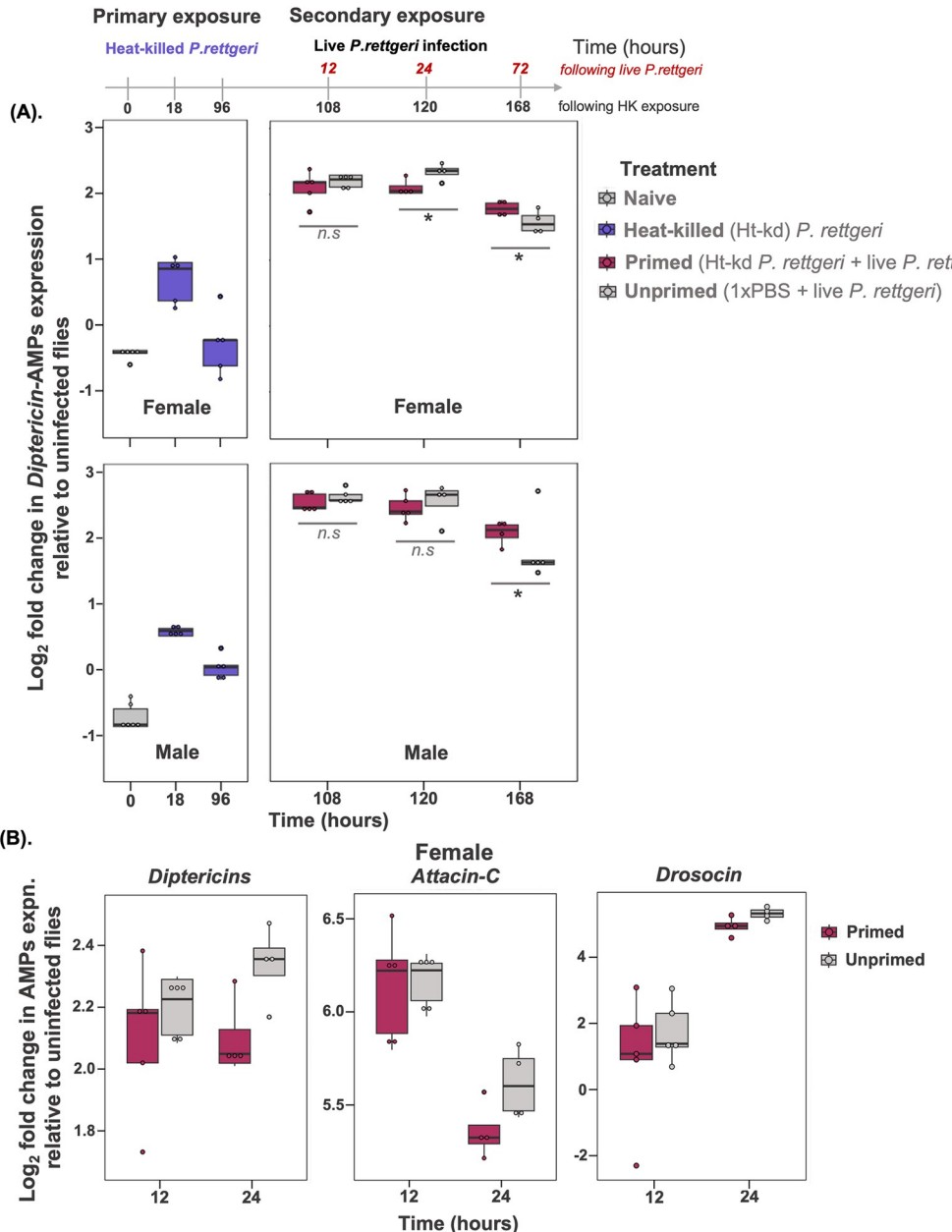

**Fig 9. Antimicrobial gene expression following priming.** (A) Mean±SE (standard error) of *Diptericin*-AMPs expression at different time points (hours) for females and males wild-type w[1118] flies—naïve or unhandled flies (time 0h); flies exposed to initial heat-killed *P. rettgeri* (blue); primed flies that are initially exposed to heat-killed *P. rettgeri* followed by challenge with *P. rettgeri* (dark red); unprimed flies that receive 1xPBS during primary exposure followed by live *P. rettgeri* during secondary exposure (grey) [n = 5 groups of 3 pooled flies per sex/treatment/timepoint]. (B). *Diptericin (A+B), Attacin-C* and *Drosocin* AMPs expression 12-hours and 24-hours after exposure to live *P. rettgeri* in w[1118] female flies. *Diptericin* data is the same as shown in panel A (top right panel), shown here for clarity and for comparison with *Attacin-C* and *Drosocin*. Asterisks '*' indicates that primed and unprimed individuals are significantly different (p<0.05). The error bars represent standard error.

clearance in males. In case of females, despite higher *Dpt* levels at 72-hours following secondary live infection in primed flies, we did not detect any difference in bacterial clearance between primed and unprimed flies. We also tested the expression pattern of other IMD-

responsive AMPs such as *Attacin-C* and *Drosocin* in females. Overall, we found that the expression of *Attacin-C* and *Drosocin* was not different between primed and unprimed females (**Fig** 9C for *AttC and Dro* and S13 **Table**).

## Regulation of IMD by PGRPs is required for immune priming

Following gram-negative bacterial infection, DAP-type peptidoglycans from gram-negative bacteria are recognised by the peptidoglycan receptors *PGRP- LC* (a transmembrane receptor) and *PGRP-LE* (a secreted isoform and an intracellular isoform), which then activate the IMD intracellular signalling cascade [36,38]. Immune regulation is achieved by several negative regulators, including *PGRP-LB*, a secreted peptidoglycan with amidase activity, that break down peptidoglycans into smaller, less immunogenetic fragments [36,38]. *PGRP-LB* has also been shown to be important for transgenerational immune priming in *Drosophila* against parasitoid wasp infection [21]. Given the role of DptB in the observed priming benefit, we therefore decided to investigate the role of *PGRPs* in the immune priming we observed during *P. rettgeri* infection.

To address this, we used fly lines with loss-of-function in *PGRP-LB*, *LC* and *LE*. Regardless of which *PGRP* was disrupted, we observed that flies were no longer able to increase their survival following an initial exposure (**Fig** 10A and S14 **Table**). However, unlike similar outcomes with *Relish* or *ΔAMP*, here the lack of priming was not driven by an inability to clear bacterial loads, as microbe loads in *PGRP* mutants were ~100-fold lower than in *Relish* or AMP mutants (**Fig** 10B and S15 **Table**). Further, *Dpt* expression compared to the $w^{1118}$ control was either similar (PGRP-LC, PGRP-LE) or higher (PGRP-LB), as expected, in both primed and unprimed flies (S5 Fig and S16 **Table**). This suggests that *Diptericin* expression is required, but not sufficient, for successful priming, which requires adequate regulation by *PGRPs*. Which specific molecular signal modifies *PGRP-mediated* regulation of *Diptericin* expression following an initial priming challenge is unclear, but must lie upstream of the IMD pathway.

## Discussion

The observation of immune priming in invertebrates underlines the importance of whole organism research in immunity in lieu of a purely mechanistic approach to immunology, highlighting that the same phenomenology can originate in very different mechanisms [18,39]. Thus, substantial experimental evidence in both lab adapted and wild-caught arthropods suggests that immune priming is a widespread phenomenon, and is predicted to have a profound impact on the outcome of host–pathogen interactions, including infection severity and pathogen transmission [40,26,27,17].

In the present work, we investigated the occurrence, generality, and mechanistic basis of immune priming in fruit flies when infected with the gram-negative pathogen *Providencia rettgeri*. We present evidence that priming with an initial non-lethal bacterial inoculum results increased survival after a secondary lethal challenge with the same live bacterial pathogen. This protective response may last at least two weeks after the initial exposure, is particularly strong in male flies, and occurs in several genetic backgrounds. We show that the increased survival of primed individuals coincides with a transient decrease in bacterial loads, and that this is likely driven by the expression of the IMD-responsive AMP *Diptericin-B* in the fat body. Further, we show that while *Diptericin* is required as the effector of bacterial clearance, it is not sufficient for immune priming, which requires the regulation by at least three PGRPs (*PGRP-LB*, *PGRP-LC*, and *PGRP-LE*). Therefore, despite having an intact IMD signalling cascade, and being able to express *Diptericin*, flies lacking any one of *PGRP-LB*, *LC* or *LE* were

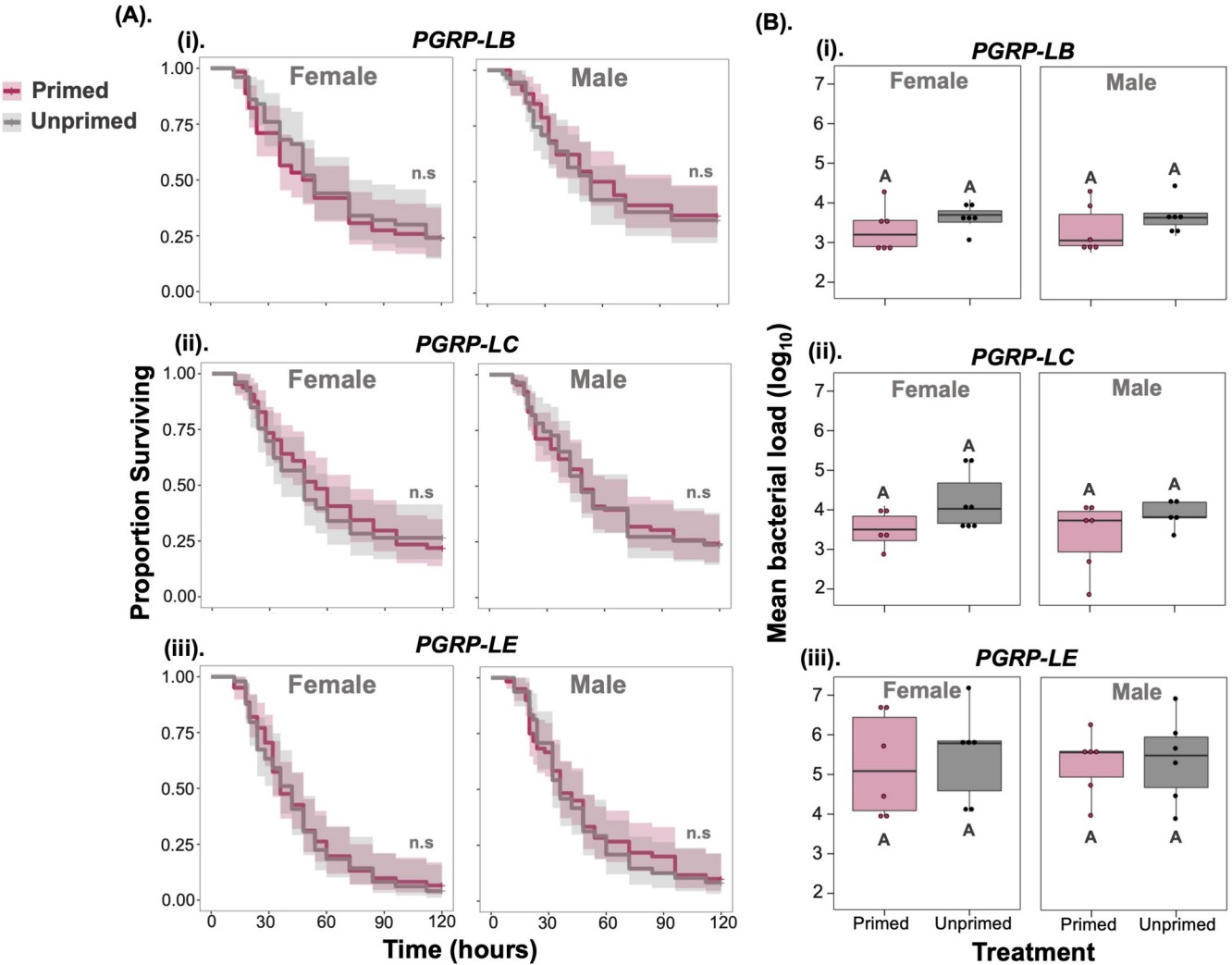

**Fig 10.** **(A).** Survival curves for males and females of Loss-of-function in *PGRPs* (peptidoglycan recognition proteins) of IMD **(i)** *PGRP-LB* **(ii)** *PGRP-LC* and **(iii)** *PGRP-LE* **(B).** internal bacterial load quantified after 24-hours post-secondary exposure in female and male flies (n = 7–9 vials with 10–15 flies in each vial/sex/treatment/ fly lines). The same letters in panel-B denote that primed and unprimed individuals are not significantly different, tested using Tukey's HSD pairwise comparisons for each fly line and sex combination. The error bars represent standard error.

not capable of increasing survival following an initial sublethal challenge. Together, our data indicates that PGRPs are necessary for regulating immune priming against *P. rettgeri*.

These results are largely consistent with other work showing that the pathways that signal pathogen clearance may not be the same that underlie the signalling of the priming response. For example, Cabrera et al [24] investigated priming with the Gram-positive *E. faecalis* and found that while the Toll pathway was required for responding to a single infection, Toll signalling was dispensable for immune priming [24]. By contrast, IMD-deficient flies were still able to clear single *E. faecalis* infections (due a functional Toll response) but were no longer able to increase survival following an initial exposure. That is, in support of our results here, the IMD played a distinct role in signalling priming that was independent of its role in clearance. Further, previous work found that the Toll pathway is required (though not sufficient) for priming in response to gram-positive bacterial *Streptococcus pneumoniae* infection [9] and here we find that the Toll pathway is not required for a functional immune priming response.

In response to infection, the expression of several AMP genes in *Drosophila* increases and then drops once the infection threat is resolved or controlled. Previous studies have shown that this occurs within a period of few hours during bacterial infections [41], and that in many cases, this increased AMP expression underlies the immune priming response [18,42]. In other host-pathogen systems, genome wide insect transcriptome studies have identified upregulation of several AMPs in primed individuals, for example, *Attacins, Defensins and Coleoptericins* in flour beetles [43,44], *Cecropin, Attacin, Gloverin, Moricin* and *Lysozyme* in silkworms [45], *Gallerimycin and Galiomicin* in wax-moths [46] and finally, *Cecropin* in tobacco moths and fruit flies [47,37]. However, there are also examples where innate immune priming involves the downregulation of AMP expression, such as priming with the gram-positive *E. faecalis* [24].

One key aspect of our results is that immune priming was not the result of continued upregulation of *Diptericin* following the initial exposure to heat-killed bacteria. Instead, we observed that the initial immune response to heat-killed bacteria had already been resolved at 72-hours, before the secondary exposure to a lethal infection at 96-hours (see **Fig** 9A). This second up-regulation of AMP expression was at least 10-times higher than the response to heat-killed bacteria, and was initially similar between both primed and unprimed flies (exposed to a sterile solution in the first exposure). It was only at 72-hours following the second lethal exposure that we observed significantly higher expression of *Diptericin* in primed flies. This difference appears to arise because unprimed flies show a faster resolution of the immune response compared to primed flies; that is, the expression of *Diptericin* shows a faster decline between 24-hours and 72-hours post-exposure in unprimed compared to primed flies (see **Fig** 9A). These patterns of gene expression provide a partial explanation for the increased survival following priming, but they do not explain why primed individuals had lower bacterial loads at 24-hours after exposure to the second lethal infection, as at this timepoint we did not detect any differences in *Diptericin* expression between primed and unprimed flies. It is also unclear why primed females showed increased expression of *Diptericin* after 24-hours, but do not show any reduction in bacterial loads at this timepoint following priming, in the way male flies did.

The sex differences we observed in priming reflect a larger pattern of sexual dimorphism in immunity present in most organisms, including *Drosophila* [48–50]. Here, we found that males showed better bacterial clearance after initial exposure to heat-killed *P. rettgeri* which enabled them to experience enhanced survival compared to females, who exhibited higher bacterial loads and greater mortality. Previous work has established several ways in which the immune responses of male and female Drosophila may differ, including Toll [51] and IMD [52] signalling or signalling of damage during enteric infections [53]. Our results indicate that the Toll pathway is not required for successful priming, so sex-differences in Toll signalling are unlikely to be important in explaining sexually dimorphic priming. However, other work has shown that disrupting the negative regulator of IMD, *PGRP-LB*, affected survival to a greater extent in females following *E. coli* infection [52]. Given that we identified a role for *PGRP*-mediated regulation of *Dpt* as key for immune priming, sex differences in *PGRP* regulation could potentially explain differences in priming between males and females.

Another aspect related to sex differences in priming, was that males experienced a survival benefit of prior exposure in every experiment we report; female flies showed more variable response, showing a survival benefit of priming repeatedly in several independent experiments (Figs 1, 2, 3, and 6) but not in others (Fig 4). Further while the priming phenotype was successfully recovered in males by restoring expression of the AMP *Dtp*, this was not observed in females. These variable outcomes in females remain puzzling, and are likely not the outcome of heritable sex-differences–it is hard to imagine how a priming response might evolve in

males, but not females, and we did repeatedly observe it. Instead, one possibility for the observed sex-differences could relate to unmeasured environmental variation in our experimental setups–although these were always minimised as much as possible–which may interact with different nutritional demands and metabolic activities in female *Drosophila*. For instance, female fruit flies are often able to reallocate resources in accordance with their reproductive demands, as observed in terminal investment during infection [54–56]. Studies from other insects suggest that inducing priming responses can directly reduce the reproductive fitness in mosquitoes [57], wax moth [58], and mealworm beetles [59]. There are therefore potential trade-offs between investment in reproductive effort and investment in stronger immune responses following priming. Given that all females used in this study were mated, it is possible that such trade-offs forced a reallocation of resources towards reproduction, thereby reducing the observed magnitude of the priming response in females. Future studies may consider comparing priming responses in females with different reproductive states in order to test whether immune priming is costlier for female *Drosophila*.

A subsidiary finding of this work was the effect of the endosymbiont *Wolbachia* on immune priming. How endosymbionts that are widespread among insects are likely to influence innate immune priming is a topic of considerable interest [60]. There is abundant evidence that *Drosophila* carrying the endosymbiont *Wolbachia* are better able to survive infections, especially viral infections [61,29,62,30]. In this case, we observed that the priming response that was present in male flies cleared of *Wolbachia* disappeared in males carrying the endosymbiont. This effect is unlikely due to a direct effect of *Wolbachia* on the ability to clear *P. rettgeri* in primed flies, as previous work has found that *Wolbachia* had no effect on the ability to suppress *P. rettgeri* during systemic infection [63]. Indeed, if priming is the result of upregulation of AMP expression, this may suggest that *Wolbachia* may be actively suppressing the expression of AMPs, thereby reducing the beneficial effects of priming. However, this hypothesis would contradict work showing that some *Wolbachia* strains upregulate the host's immune response and result in a reduction of pathogen growth [64,65]. Another possibility is that while both *Wolbachia* and *P. rettgeri* are Gram-negative bacteria, they could elicit the expression of competing AMP responses, leading to a less efficient priming response. Further rigorous experimental work is therefore required to fully understand this effect of *Wolbachia* on immune priming.

Finally, it is important to consider the implications of immune priming for disease ecology and epidemiology, and particularly how it may affect pathogen transmission. If priming acts by improving bacterial clearance via increased AMP expression, as observed in the present work, we predict that priming is likely to reduce pathogen shedding at the individual level, resulting in reduced disease transmission at the population level. Epidemiological models that have incorporated priming predict that primed individuals with enhanced survival following an initial sub-lethal pathogenic exposure are less likely to become infectious upon re-infections [27]. It remains unclear whether immune priming reduces pathogen transmissibility, varies the infectious period, or alters infection-induced behavioural changes in the host [27]. An additional level of complexity is that there is likely to be substantial within-population genetic variation in how priming affects each of these components of pathogen transmission. Insects offer a powerful system to investigate these effects, because they rely on an innate immune system that can induce an easily measurable priming response, but further, insect are also important vectors of many infectious diseases. Immune priming has thus emerged as a provocative idea to reduce the vectorial capacity of insect vectors, thereby reducing the transmission of vector-borne pathogens [17]. A better understanding of how immune priming contributes to host heterogeneity in disease outcomes would aid our understanding of the causes and consequences of variation in infectious disease dynamics.

## Materials and methods

### Fly strains and maintenance

Several *D. melanogaster* strains: $w^{1118}$ (Vienna *Drosophila* Resource Center), *iso-w$^{1118}$* (Bloomington *Drosophila* Stock Center), Canton-S, Oregon-R *Wolbachia$^{+ve}$ and Oregon-R Wolbachia$^{-ve}$* [30]. Transgenic flies included immune mutants $Rel^{E20}$ (relish—IMD pathway regulator [66]) and *spz (spätzle– Toll* pathway regulator [67]), and were a gift from the Saleh lab (Pasteur, Paris). We also used the following CRISPR/Cas9 deletion lines, originally gifted by the Lematire lab (EPFL, Lausanne) [34]: (a) ΔAMPs—flies lacking 10 fly AMPs, (b) Group-B—flies lacking major IMD regulated AMPs including *Attacins (AttC$^{Mi}$; AttD$^{SK1}$), Drosocin (Dro$^{SK4}$)* and *Diptericins (Dpt$^{SK12}$)*, (c) $Dpt^{SK12}$ –flies lacking *Diptericins (DptA and DptB)*, and (d) ΔAMPs$^{+Dpt}$–flies lacking 10 known AMPs except *Diptericins*. All the CRISPR/Cas9 mutants were generated previously from the *iso-w$^{1118}$* genetic background using CRISPR/Cas9 gene editing technology to induce null mutations in the selected genes [34].

We also used the binary GAL4/UAS system for tissue specific silencing of target genes with the RNA interference (RNAi) method. *Diptericin-B (DptB)* (Bloomington stock# *28975*) was tissue-specifically knocked down in the fat body (*w$^{1118-iso}$,Fb-Gal4i+(P{fat})*) and in haemocytes [*w$^{1118-iso}$,Hmldelta- Gal4; He-Gal4* –a combination of two haemocyte GAL4 drivers, the Hml-GAL4.Δ [68] and He-GAL4.Z [69]]. All fly stocks and experimental flies were maintained at 25˚C ±1˚C on a 12:12 hour light: dark cycle in vials containing 7ml of standard cornmeal fly medium [70]. For the experiments, we controlled larval density by placing 10 females and 5 males in each vial and the females were allowed to lay eggs for 48-hours. Fourteen days later, the eclosing males and females were sorted and collected and separated into group of 25 flies in each vial. Three-day old, mated individuals were used in all experiments.

### Systemic immune priming and infection assays

*P. rettgeri* was grown at 37˚C in 10ml Luria broth (Sigma Ltd) overnight to reach optical density $OD_{600}$ = 0.95 (measured at 600nm in a Cell Density Meter, Fisherbrand). The culture was centrifuged at 5000 rpm for 5 min at 4˚C, and the supernatant was removed, and the final OD was adjusted to 0.1 and 0.2 by using sterile 1xPBS (Phosphate buffer saline). To obtain heat-killed bacteria the dilution was incubated at 90˚C for 20–30 mins [11]. To ensure all bacteria were dead in the heat-treated culture, it was plated and no growth was confirmed. To prime individuals, 3-day old adults were pricked with a 0.14-mm pin (Fine Science Tools) dipped in either heat-killed bacteria for the primed treatment or in 1xPBS solution for the unprimed treatment (exposed to a sterile solution in the first exposure) in the mesopleuron region (the area situated under the wing and to the left of the pleural suture) [71]. Following this initial priming treatment, the individuals were pricked using $OD_{600}$ = 0.1 live *P. rettgeri* bacteria (resulting in approximately 70 bacterial cells/fly). To test whether male and female adult flies show priming (measured as enhanced survival) with increasing time intervals between the initial heat-killed exposure and later challenge with live *P. rettgeri*, we tested several time points between the two challenges 18-hours, 48-hours, 96-hours, 1-week and 2-weeks (See S1 **Fig** for experimental design; n = 9–13 vial treatment/sex/ fly line). We used a split vial experimental design to obtain replicate matched data for both survival and bacterial load see [72] for details. Briefly after infection each vial containing about 25 flies (of each treatment, sex and fly line combination) were divided into 2 vials for measuring (i). survival following infection (see S1i **Fig**; 13–17 flies/combination) and (ii). internal bacterial load (see S1ii **Fig**).

### Bacterial load quantification

To test whether the host's ability to supress bacterial growth varies across primed and unprimed individuals, we quantified bacterial load as colony forming units (CFUs) at

24-hours after *P. rettgeri* infection for controls and transgenic flies in both sexes. Flies were surface sterilized in groups of 3–5 flies per vial in 70% ethanol for 30s and washed twice with distilled water before homogenising flies individually using micro pestles. We immediately performed serial dilutions of the homogenate with 1xPBS and plated them on LB agar plates and cultured at 29°C overnight. The following day, we counted the CFUs manually [73] (n = 9–13 vial/sex/treatment/fly mutants).

## Gene expression quantification

The expression of *Diptericins* was quantified by qRT-PCR. In parallel with the survival experiment, we randomly selected a subset of control $w^{1118}$ individuals (both males and females) for RNA extraction, we included 15 flies [5–7 replicates of 3 flies pooled together for each treatment (primed and unprimed) for both males and females]. We randomly removed selected flies (3 flies per vial) at different time points post exposure to *P. rettgeri* (18-hours and 72-hours post-priming, 12-hours and 24-hours post-challenge). We then homogenised pools of three in 80µl of TRIzol reagent (Invitrogen, Life Technologies). Homogenates were kept frozen at -70°C until RNA extraction. We performed mRNA extractions using the standard phenol-chloroform method and included a DNase treatment (Ambion, Life Technologies).

We confirmed the purity of eluted RNA using a Nanodrop 1000 Spectrophotometer (version 3.8.1) before going ahead with reverse transcription (RT). The cDNA was synthesized from 2µl of the eluted RNA using M-MLV reverse transcriptase (Promega) and random hexamer primers, and then diluted 1: 1 in nuclease free water. We then performed quantitative RT-PCR (qRT-PCR) on an Applied Biosystems StepOnePlus machine using Fast SYBR Green Master Mix (Invitrogen) using a 10µl reaction containing 1.5L of 1:1 diluted cDNA, 5µl of Fast SYBR Green Master Mix an 3.5µl of a primer stock containing both forward and reverse primer at 1µM suspended in nuclease free water (final reaction concentration of each primer 0.35µM). For each cDNA sample, we performed two technical replicates for each set of primers and the average threshold cycle (Ct) was used for analysis. We obtained the *AMP* primers from Sigma-Aldrich Ltd; *Dpt*_Forward: 5' GACGCCACGAGATTGGACTG 3', Dpt_Reverse: 5' CCCACTTTCCAGCTCGGTTC 3', *AttC*_Forward: TGCCCGATTGGACCTAAGC, AttC_Reverse: GCGTATGGGTTTTGGTCAGTTC, *Dro*_Forward: ACTGGCCATCGAGGAT CACC, *Dro*_Reverse: TCTCCGCGGTATGCACACAT. We used *RpL*49 as endogenous reference gene, *RpL*49_Forward: 5' ATGCTAAGCTGTCGCACAAATG 3', *RpL*49_Reverse: 5' GTTCGATCCGTAACCGATGT 3'. We optimised the annealing temperature ($T_a$) and the efficiency (Eff) of the *Dpt* primer pair was calculated by 10-fold serial dilution of a target template (each dilution was assayed in duplicate); *Dpt*: $T_a$ = 59°C, Eff = 102%; *AttC*: $T_a$ = 60°C, Eff = 94%; *Dro*: $T_a$ = 61°C, Eff = 104%. We analysed the gene expression data by calculating the ΔΔCT value [74]. Fold change = $2^{-\Delta\Delta Ct}$

Where, ΔΔCt = [(Ct of Gene A–Ct of Internal control) of Infected sample]–

[(Ct of Gene A–Ct of Internal control) of Control sample]

We used *RpL32* as a reference gene as it was expressing steadily in our treatment and control conditions. We calculated fold change in gene expression relative to the uninfected controls to calculate ΔΔCT and used ANOVA to test whether AMP expression differed significantly between primed and unprimed treatment for males and females.

## Oral priming and infection

For oral priming and live infection, we adjusted the final concentration to $OD_{600}$ = 25 [53,73]. We initially prepared vials for oral priming by pipetting 350–400 µl of standard agar [see [73]] onto lid of a 7ml tubes (bijou vials) and allowed it to dry. Simultaneously, we starved the

experimental flies on 12ml agar vials for 4 hours. Once the agar on the lids dried, we placed a small filter disc (Whattmann-10) in the lid and pipetted 80μl of heat-killed bacterial culture (primed treatment) or 5% sucrose solution (unprimed control treatment) directly onto the filter disc. Once the agar dried, we orally exposed flies (heat-killed only, primed and unprimed treatment) by adding approximately 10 flies per vial for 18-hours and then transferred the flies onto fresh Lewis food vials. After 3-days, we again prepared the bijou vials and once the agar dried, this time we added 80μl of live bacterial culture ($OD_{600}$ = 25) and exposed flies to live *P. rettgeri* for 18-hours. We then transferred flies onto fresh food vials and observed survival after oral exposure to *P. rettgeri* every 12 hours for the following 8 days.

## Measuring locomotor activity

We measured the locomotor activity of single flies (n = 52 flies for each for each sex and treatment combination) during three continuous days using a *Drosophila* Activity Monitor–DAM (v2 and v5) System [75], in an insect incubator maintained at 25˚C ± 1˚C in a 12 D: 12 L cycle. We then processed the raw activity data using the DAM System File Scan Software [75]. We analysed fly activity data using three metrics [76–79]: total activity, the average activity during 5-min activity bouts, and proportion of 5-min bouts with zero activity (which has been defined as sleep in *Drosophila* [80]) (**S2III Fig**)

## Measuring bacterial shedding

We measured the bacterial shedding of single flies (n = 8–12 flies per treatment and sex combination) at a single time point, 4-hours following overnight oral bacterial exposure. We chose this timepoint as in other work we have found that most faecal shedding of bacterial pathogens occurs within the first 4 hours, and steadily decreases by 8 hours following overnight oral infection. Following oral priming to either heat-killed bacterial culture (primed treatment) or 5% sucrose solution (unprimed treatment), flies were exposed to an oral infection with live infection with 80μl of live *P. rettgeri* culture ($OD_{600}$ = 25). Following 18 hours of oral exposure, flies were placed individually into 1.5ml Eppendorf tubes with approximately 50μl of Lewis medium in the bottom of the tube for 4-hours. After 4-hours, flies were removed from the tube and the remaining content of each tube was washed with 50μl of 1xPBS buffer by vortexing thoroughly for at least 5 secs. We then plated these samples on a LB agar plates, incubated them at 29˚C and counted the colonies manually after 18-hours.

## Measuring bacterial transmission

We measured transmission in groups of flies, by collecting age-matched donor and recipient *w[1118]* flies, separately for each sex. To test the effect of priming on the ability of flies to transmit *P.rettgeri*, we orally exposed 3-day old *w[1118]* flies (donor flies) with either 5% sucrose and heat-killed bacteria (primed) or 5% sucrose only (unprimed). After 96-hours the donor flies were exposed to $OD_{600}$ = 25 of *P. rettgeri* (*see oral infection section above*). The infected donors were marked by cutting the corner of a fly wing. We then placed one donor and five uninfected recipient flies in 7ml bijou vials with a small amount of Lewis food on the lid for each treatment (heat-killed bacteria only and 5% sucrose exposed without live infection) and sex-combination. After 4-hours exposure we surface-sterilised and homogenized the flies and plated the homogenate to measure the presence or absence of *P. rettgeri* infection inside each recipient fly, as an indication of successful transmission of bacteria from the donors to the recipient flies. We also set up and plated flies in groups with no infection, to confirm that our measures of *P. rettgeri* prevalence reflected successful transmission from the donor flies.

## Data analysis

All statistical analyses and graphics were carried out and produced in R (version 4.2.2) using the ggplot2, coxhz and lme4 packages [81–83], and all raw data and code is available at 10.5281/zenodo.7624084 [84] or in S1 Data. We analysed the survival data following systemic *P. rettgeri* infection with a mixed effects Cox model using the R package 'coxme' [82] for different treatment groups (that is, primed and unprimed) across both the sexes and fly lines (controls and transgenic lines). We specified the model as: survival ~ treatment * sex * (1|vials/block), with 'treatment' and 'sex' and their interactions as fixed effects, and 'vials' nested within each 'block' as a random effect for control and transgenic lines. We used ANOVA to test the impact of each fixed effect in the 'coxme' model. We analysed the bacterial load, measured as $log_{10}$ bacterial colony-forming units (CFUs) at 24-hours following *P. rettgeri* infection. As bacterial load data was non-normally distributed, we log-transformed the data and analysed using a non-parametric one-way ANOVA (Kruskal-Wallis test) to test whether the 'treatment' groups that is, primed and unprimed individuals significantly differed in internal bacterial load for males and females of each fly lines (control and transgenic lines). Data for bacterial shedding and transmission were non-normally distributed (tested using Shapiro-Wilks test for normality) hence we performed non-parametric Wilcoxon Kruskal-Wallis tests for bacterial shedding and transmission data.

## Supporting information

**S1 Data. All data used to generate figures.**
(XLSX)

**S1 Fig.** Schematic representation of different priming experiments aimed at (I). testing whether the length of the period between primary heat-killed exposure and the secondary pathogenic challenge affects the extent of priming (II). how different lab-adapted control/genetic background flies vary in priming (III). dissecting the role of innate immune pathways (IMD and Toll) and inducible AMPs in immune priming and (IV). deciphering mechanisms that bring about immune priming in Drosophila using tissue-specific fat body and haemocytes UASRNAi mutants. The experimental design for priming assays includes survival and internal bacterial load quantification.
(TIFF)

**S2 Fig. Experimental design to measure epidemiological components following systemic (OD600 = 1) and oral (OD600 = 25) priming and infection with initial heat-killed exposure followed by live P. rettgeri in male and female wildtype w1118 flies.** The assays include (I). survival following different infection routes (II). internal bacterial load (III). behavioural components of pathogen exposure such as sleep and awake activity (IV). bacteria shedding and (V). transmission. n = 6–7 vials of 8–12 flies in each vial, for each treatment and sex combination.
(TIFF)

**S3 Fig. The effect of priming on fly locomotor activity.** Mean ±SE total locomotor activity for males and females (n = 52 individual flies per treatment), during first 72-hours following systemic and oral priming and infection. (A) average total locomotor activity (B) average awake activity and (C) proportion of flies spent sleeping.
(TIFF)

**S4 Fig. Survival curves of w1118 and iso-w1118 (drosdel).** flies after initial heat-killed exposure and followed by live P. rettgeri infection with OD600 = 0.1. As another control, we

infected both w1118 and iso-w1118 control because the CRISPR/cas9 AMP mutants we used were on the iso-w1118 background, so we wanted to confirm that any changes in priming were not due to the background of the mutants, as opposed to the mutations. We found that the differences between w1118 and iso-w1118 (primed and unprimed treatments) were not significantly different.
(TIFF)

**S5 Fig. Diptericin expression in PGRP deletion lines.** 24-hours after exposure to live P. rettgeri in male and female control w1118 flies and flies with loss-of-function in different PGRPs, PGRP-LB, PGRP-LC & -LE.
(TIFF)

**S1 Table. Summary of mixed effects Cox model, fitting the model to estimate time-delayed priming response in control w$^{1118}$ male and female flies.**
(DOCX)

**S2 Table. Summary of mixed effects Cox model, fitting the model to estimate priming response in different laboratory control w$^{1118}$ male and female flies.**
(DOCX)

**S3 Table. Summary of mixed effects Cox model, fitting the model to estimate the impact Wolbachia on immune priming response using genetic background OreR male and female flies.**
(DOCX)

**S4 Table. Summary of mixed effects Cox model, fitting the model to estimate priming response in male and female control w$^{1118}$ flies.**
(DOCX)

**S5 Table. Summary of log$_{10}$ transformed bacterial load.**
(DOCX)

**S6 Table. Summary of Cox prop-hazard model, for female and male flies of wild type w$^{1118}$.**
(DOCX)

**S7 Table. Summary of log transformed bacterial load data.**
(DOCX)

**S8 Table. Model outputs for statistical test (GLM) performed on host activity data.**
(DOCX)

**S9 Table. Summary of non-parametric Wilcoxon (Kruskal-Wallis) test for oral bacterial shedding (log transformed bacterial load).**
(DOCX)

**S10 Table. Summary of mixed effects Cox model, fitting the model to estimate priming response in male and female control w$^{1118}$, IMD and toll transgenic flies.**
(DOCX)

**S11 Table. Summary of log$_{10}$ transformed bacterial load data in amp deletion lines.**
(DOCX)

**S12 Table. Summary of mixed effects Cox prop-hazard in tissue-specific Dpt knockdown lines.**
(DOCX)

**S13 Table. Summary of log$_{10}$ transformed Dpt gene expression data.**
(DOCX)

**S14 Table. Summary of mixed effects Cox model, fitting the model to estimate priming response in male and female PGRP mutant flies.**
(DOCX)

**S15 Table. Summary of log$_{10}$ transformed bacterial load data after *P. rettgeri* infection.**
(DOCX)

**S16 Table. Summary of log$_{10}$ transformed Dpt gene expression data in flies with different PGRP deletion transgenic flies.**
(DOCX)

## Acknowledgments

We thank Bruno Lemaitre's lab for generating and generously sharing the CRISPR/Cas9 AMP transgenic lines. We thank Emily Robertshaw, Srijan Seal, Saubhik Sarkar and Pavan Thunga for laboratory assistance. We thank Sveta Chakrabarti and Ashworth fly group members for helpful discussion. Finally, we thank Angela Reid, Lucinda Rowe, James King and Alison Fulton for help with media preparation.

## Author Contributions

**Conceptualization:** Arun Prakash, Pedro F. Vale.

**Data curation:** Arun Prakash.

**Formal analysis:** Arun Prakash, Florence Fenner.

**Funding acquisition:** Pedro F. Vale.

**Investigation:** Arun Prakash, Florence Fenner, Biswajit Shit, Tiina S. Salminen, Katy M. Monteith.

**Methodology:** Arun Prakash, Florence Fenner, Biswajit Shit, Tiina S. Salminen, Katy M. Monteith.

**Project administration:** Pedro F. Vale.

**Resources:** Imroze Khan, Pedro F. Vale.

**Supervision:** Imroze Khan, Pedro F. Vale.

**Visualization:** Arun Prakash.

**Writing – original draft:** Arun Prakash, Pedro F. Vale.

**Writing – review & editing:** Arun Prakash, Florence Fenner, Biswajit Shit, Tiina S. Salminen, Katy M. Monteith, Imroze Khan.

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
