## [Decision Letter · Decision Letter 0]

22 May 2024

Dear Vale,

Thank you very much for submitting your manuscript "IMD-mediated innate immune priming increases Drosophila survival and reduces pathogen transmission" for consideration at PLOS Pathogens. As with all papers reviewed by the journal, your manuscript was reviewed by members of the editorial board and by several independent reviewers. In light of the reviews (below this email), we would like to invite the resubmission of a significantly-revised version that takes into account the reviewers' comments.

Both reviewers found this to be an important topic and the results were novel and interesting. As well as a number of changes to the text, reviewer 2 questions whether some controls are missing and suggests removing these experiments. This needs careful attention in the revised version. They also suggest that the manuscript is generally simplified to increase its impact.

We cannot make any decision about publication until we have seen the revised manuscript and your response to the reviewers' comments. Your revised manuscript is also likely to be sent to reviewers for further evaluation.

Sincerely,

Francis Michael Jiggins

Section Editor

PLOS Pathogens

Francis Jiggins

Section Editor

PLOS Pathogens

Michael Malim

Editor-in-Chief

PLOS Pathogens

orcid.org/0000-0002-7699-2064

Both reviewers found this to be an important topic and the results were novel and interesting. As well as a number of changes to the text, reviewer 2 questions whether some controls are missing and suggests removing these experiments. This needs careful attention in the revised version. They also suggest that the manuscript is generally simplified, which may increase its impact.

Reviewer's Responses to Questions

**Part I - Summary**

Reviewer #1: This study aims to clarify the phenomena of immune priming in invertebrates, which in my opinion is of big importance for several fields (e.g., epidemiology, conservation) and has been heavily studied but in a rather descriptive manner. The authors instead tackled the mechanistic part of the phenomena - very poorly studied in past studies - which I appreciated and think it contributes substantially to the field. To my knowledge, there are only a couple of old studies that demonstrate immune priming in Drosophila with bacterial infections. Wolbachia-presence/absence in these studies is also often neglected, and that is not the case here, which I again appreciated. So, it is quite interesting, and might I dare to say "novel", that this phenomenon has been found most recently in Drosophila and has been fairly well described in response to this particular bacterial infection. I would also like to add the importance of the standpoint the authors take in this study: acknowledging the complexity of infection and how the latter influences shedding and locomotor activity (i.e., potential proxy for contact rate) and therefore, pathogen transmission, which in turn is crucial and not often observed in the field. I believe this is rather relevant nowadays and contributes to the understanding of disease evolution and spread in other model systems besides the fruit fly and is well emphasized in the last paragraph of the discussion.

The manuscript is quite well-written, straightforward and easy to follow (in particular the results were a delight to read even when they are this dense). The experiments are well design and also quite easy to follow, making a rather complete story in the end. The authors not only observed, expanded, tested for specificity and duration of the immune priming, but also measured its consequences for transmission and the mechanisms underpinning the phenomena with good methodologies. They also were clear about their intentions and the limitations of such methodologies (for example how phagocytosis and melanisation might also play a role). The methods are fairly well-described with the standard nomenclature and I do not think additional work is necessary and would actually harm the study by undermining some results.

I think this study is of great importance for the field and for the readers of PLOS Pathogens. Moreover, it aligns with the journal interests, and so, I would mostly the authors to address the overall minor comments to the manuscript.

Reviewer #2: This manuscript summarizes many different aspects of immune priming in Drosophila melanogaster against the pathogenic Gram-negative bacterium Providencia rettgeri. The authors characterize the lifespan of the immune priming, its sex and pathogen specificity as well as septic and oral routes of exposure, before investigating more closely the role of different immune pathways and antimicrobial peptides in mounting a successful priming response. They demonstrate that the IMD pathway and the AMP Diptericin are indispensable for successful immune priming against this bacterial pathogen. Finally, they report on transmission rates being influenced by the immune history of an individual and conclude how immune priming can have profound epidemiological consequences.

This study presents a very thorough investigation of many diverse aspects of immune priming, which is a great contribution to the field and broadens our understanding of immune memory and its consequences. However, I could not help but notice a general lack of focus. Extremely important and novel findings are hidden between other experiments that seem rather superficial or not relevant to the story the authors want to tell. This was quite distracting while reading and a more compact version might be more appropriate.

**Part II – Major Issues: Key Experiments Required for Acceptance**

Reviewer #1: I think the discussion could benefit of a bit more explaining, although I understand the difficulty in doing so with this number of results. However, here are some examples:

• sexual dimorphism in fruit flies has been associated to differences in ROS expression in the gut and other tissues, what might explain in part of the sexual dimorphic results we observe here.

• Wolbachia is a gram-negative bacterium but might still induce different immune effectors, that trade-off with the ones needed for Providencia.

• The lack of locomotor activity could be better discussed and, in my opinion, might reflect the transmission route (which is briefly mentioned in the results) of the pathogen, that seems to take a more sit and wait approach.

• In the transmission experiment, you exposed them for 4 hours to the infected fly and then immediately homogenised the individuals. I wonder if that time is enough to measure transmission and if maybe you should have individualized the recipient flies, waited a bit longer and then measured the presence/absence of transmission. Particularly because I am not aware of your CFU detection threshold.

Reviewer #2: -

**Part III – Minor Issues: Editorial and Data Presentation Modifications**

Reviewer #1: Minor comments:

• Given that the results come before the methods, I would suggest paying attention to some important nuances of the methods, like the infection route used in some of the experiments, which is only explicitly mentioned later on.

• References are not consistently formatted throughout the manuscript. For instance, introduction is numeric, but in line 154 is not.

Reviewer #2: I would recommend streamlining the manuscript by removing some experiments that are lacking controls and by reducing the number of figures. The experiment on the species specificity does not consider the potential differences in virulence between the different bacteria taxa. To ensure that the priming is really species specific the priming-infection set up should have been fully reciprocal, also using the other bacteria for the priming. Or at least the exposure of an unprimed control group should have been tested with the other bacteria. Only the difference in survival between an unprimed and primed group will let us estimate a priming benefit. (Here, also the hazard ratios in Figure 3 B and C are shown twice, the only difference being the chosen reference.) The experiment regarding the locomotor activity is currently only described very briefly in the text and the different aspects of Figure 7 are not discussed in any detail. Thus, this figure might be more suited for the supplement. My concern in the experiments using the AMP mutants (Figure 9 Ai-iii) is that these have a really high susceptibility to infection with almost all flies dying, which makes it a lot more difficult to detect a priming effect. On the other hand, the finding that the priming effect can be rescued by the expression of Diptericin in the males but not females is highly interesting and should be given more importance. The same is true for the tissue specific Diptericin knockdowns with their quite exciting results (which also tend to get lost in this long list of separate experiments).

Lastly, the authors describe that their use Cox mixed effect models testing not only for the effect of priming treatment but also for differences between the sexes, however the results of the tests for sex differences are not discussed.

Minor points:

The intro could mention more examples of priming from other insects/invertebrates (besides Drosophila and woodlouse)

Figure 1: why is there no challenge control (heat killed bacteria) in the later time points

How strong is the effect of the vials (depending on how sick individuals become might affect the rest of the cohort)?

Figure 2: Is Fig2a the same as Fig1c?

L141 How the labels are referenced in the text does not seem correct. 2C should be 2D, l140 2B should be 2 C

Figure 5: There is also no priming via the septic route in the females. However, this was observed in previous experiments (figure 1). Thus, the results for the females don’t seem repeatable here and should not be attributed to the priming/infection route. The authors should either repeat this experiment or discuss the lack of septic priming in the females in this part of the study.

Figure 6B: Would it be possible to add individual data points here?

Figure 7: Is barely mentioned in the text. If the results have this little importance, the figure should be moved to the supplements.

Figure 8: The Relish mutant seems to have no resistance at all. So it would be rather difficult to detect a priming effect if in an unprimed state no flies survive the infection.

Figure11 A): the dots should not be connected as they are not repeated measures of the same fly. Also, it is not clear what the reference is? Is it the mean of the three time points for uninfected flies?

Line 393 the wrong figure is cited here, while the actual figure 12 is not introduced in the text before its appearance.

PLOS authors have the option to publish the peer review history of their article (what does this mean?). If published, this will include your full peer review and any attached files.

Reviewer #1: No

Reviewer #2: No
---

## [Editor Report · Decision Letter 1]

31 May 2024

Dear Dr Vale,

We are pleased to inform you that your manuscript 'IMD-mediated innate immune priming increases Drosophila survival and reduces pathogen transmission' has been provisionally accepted for publication in PLOS Pathogens.

Best regards,

Francis Michael Jiggins

Section Editor

PLOS Pathogens

Francis Jiggins

Section Editor

PLOS Pathogens

Michael Malim

Editor-in-Chief

PLOS Pathogens

orcid.org/0000-0002-7699-2064

The revisions address all the revewer's comments. In particular, the removal of some material has made the work more focussed and robust. Overall, this is an important contribution to the field of immune priming.

---

## [Editor Report · Acceptance letter]

5 Jun 2024

Dear Dr Vale,

We are delighted to inform you that your manuscript, "IMD-mediated innate immune priming increases Drosophila survival and reduces pathogen transmission," has been formally accepted for publication in PLOS Pathogens.

Best regards,

Michael Malim

Editor-in-Chief

PLOS Pathogens

orcid.org/0000-0002-7699-2064